# Graphene oxide enabled long-term enzymatic transesterification in an anhydrous gas flux

Weina Xu [1], Zhongwang Fu[1], Gong Chen [1], Zheyu Wang[1], Yupei Jian[1], Yifei Zhang [2], Guoqiang Jiang[1], Diannan Lu [1], Jianzhong Wu[3] & Zheng Liu [1,4]

Gas-phase enzymatic catalysis has been long pursued but not yet utilized in industrial processes due to many limitations. Herein, we report a hydroxyl-rich graphene oxide (GO) aerogel that can preserve the enzymatic activity and stability in an anhydrous gas flow by providing a water-like microenvironment. Lipase immobilized in the GO aerogel exhibits a 5 to 10-fold increase in apparent activity than the lyophilized lipase powder in transesterification of geraniol and vinyl acetate in the gas phase and maintains the initial activity for more than 500 h. The solid-state circular dichroism measurement confirms that the lipase keeps its native conformation in the aerogel, and the thermogravimetric analysis shows that water molecules essential for the lipase activity can be replaced by the hydroxyl groups at the GO surface. The versatility of this method is demonstrated for two other lipases with different structures, promising unprecedented applications of enzyme-GO aerogels to gas-phase enzymatic catalysis.

[1] Key Lab of Industrial Biocatalysis, Ministry of Education, Department of Chemical Engineering, Tsinghua University, 100084 Beijing, China. [2] Department of Biomedical Engineering, Columbia University, New York, NY 10027, USA. [3] Department of Chemical and Environmental Engineering, University of California, Riverside, CA 92521, USA. [4] Jiangsu National Synergistic Innovation Centre for Advanced Materials, 211800 Nanjing, China. Correspondence and requests for materials should be addressed to Z.L. (email: liuzheng@mail.tsinghua.edu.cn)

Harnessing the power of enzymes for applications in unnatural scenarios is challenging but tempting. Great efforts including chemical modification and genetic engineering have been made in the past decades to bring the enzymatic catalysis from aqueous solutions to non-aqueous media[1–4], stimulating revolutionary developments in the production of pharmaceuticals, biofuels, and a variety of fine chemical products[5–8]. In addition to liquid- phase reactions, enzymes can also catalyze gas-phase reactions. Nature has shown the great ability of enzymes to processing gaseous substrates. For example, ribulose-1,5-bisphosphate carboxylase/oxygenase (RuBisCo) can capture $CO_2$ for photosynthesis[9], and dinitrogenase can reduce atmospheric nitrogen to ammonia[10]. However, the development of gas-phase enzymatic catalysis has been very slow since the earliest study by Inokuchi et al. in 1969[11] and has never been used in industrial practice.

From the perspective of chemical industry and engineering, gas-phase enzymatic catalysis has many unique advantages compared with conventional liquid-phase enzymatic reactions, such as the faster diffusion of reactants, higher stability of enzymes, a greener process without using solvents, and the ease of recovery and reuse of immobilized enzymes[12,13]. In addition, gas-phase enzymatic reactions held irreplaceable merits in the processing of gaseous substrates. For instance, the detection and detoxification of nerve agents, the decontamination of volatile pollutants, and the biosynthesis of ammonia from $N_2$[14,15]. Pioneering studies have shown that enzymes (e.g. hydrogenase[11,16], alcohol oxidase[17,18], and lipase[19–22]) in their lyophilized states can react with their gaseous substrates in the gas phase at an ultralow activity (~1% of the activity in an aqueous solution). This is one of the major barriers for the industrial practice of gas-phase enzymatic catalysis. Conventional immobilization strategy has been applied to improve the activity of enzymes in the gas phase[23–25], but the improvement is too little to compete with their soluble counterparts due to the structural rigidity of the dried enzymes. Another key factor in gas-phase enzymatic reaction is the water activity ($A_w$), the ratio of the partial pressure of water in the atmosphere to that of the water saturation pressure at the system temperature[13]. A certain degree of $A_w$ (usually at a $A_w$ of 0.4 or even higher)[26,27] is required to maintain the activity of enzymes, which is usually achieved by continuously feeding water vapor or some salt solutions to compensate the stripping of water molecules in environment or on enzymes[12,28]. A fact is that the lyophilized enzyme in a rigid conformation exhibits poor activity but considerable stability, while the presence of free water molecules can soften the structures, thus leading to better catalytic activity but a higher probability to deactivation[16,29,30]. The dual role of water molecules indicates a dilemma in the gas-phase enzymatic catalysis, therefore a subtle control of the water activity is necessary. Many particular reactions may be very sensitive to the presence of moisture (e.g. hydrolysis of ester occurs at an $A_w$ of 0.025)[26], which also limits the application of gas-phase enzymatic catalysis.

Recent advances have suggested that engineering a favorable microenvironment for enzymes is an effective approach to optimizing enzymatic reactions[31,32]. Efforts have been made to rationally tailor the stability[33], activity-pH profiles[34], substrate affinity[35], and reaction specificity of enzymes in aqueous phase[36]. A very recent gas-phase example by Badieyan et al.[37] suggested that a hydrophilic microenvironment created by polymer network can preserve the activity of enzymes in a water-free condition. Compared with the lyophilized enzyme, they found that haloalkane dehalogenase immobilized on the surface of glass beads displaying poly(sorbitol methacrylate) could result in a 40-fold increase in activity at $A_w = 0.5$. They concluded that the poly(sorbitol methacrylate) replaced protein–water interactions in a water-free microenvironment.

Here we go one step further and show that a hydroxyl-rich microenvironment can even drive enzymes for catalyzing water-sensitive reactions in an anhydrous gas-phase condition. We create a hydrophilic, flexible and porous matrix which can form multiple hydrogen bonds with enzymes. In an essentially water-free environment, such a matrix would allow the necessary conformational transitions of enzymes, but avoid the detrimental collision by water molecules and enable a long-term enzymatic catalysis.

## Results and discussion

**Preparation and characterization.** Graphene oxide (GO) is a flexible sheet capable of forming hydrogels with biomolecules, such as enzymes, via various interactions including hydrogen bonding, coordination, electrostatic interaction, and π–π stacking[38,39]. In a recent review, Stamatis and coworkers[40] summarized various approaches to functionalizing GO for enzyme immobilization and discussed the practical potential of GO-immobilized enzyme for aqueous synthesis. Great efforts have been directed at immobilizing lipase on GO in liquid media. For example, Sofer and coworkers[41,42] studied lipases from *Rhizopus oryzae*, *Candida rugosa*, and *Penicillium camemberti* and found that GO enhanced the thermal stability, solvent tolerance and activity of the enzyme in organic solvents. Husain and coworkers[43] attached lipase from *Aspergillus niger* onto polyaniline-coated silver-functionalized GO nanocomposites and achieved an improved thermal stability and activity in aqueous solutions. Improved stability in organic solvents was also reported for lipases from *Candida rugosa* and *Candida sp.* immobilization on modified GO sheets[44–47]. In addition, immobilization on GO-contained magnetic supports improves the stability of *Porcine pancreas* lipase[48–50], and *Candida rugosa* lipase[51]. We appreciate the richness of hydrophilic functional groups on GO sheets[52], because it enables enzymes to stay in a hydrophilic and structurally flexible network, where the enzymes can maintain their native conformation in an anhydrate condition. Here, we choose *Candida Antarctica* lipase B (CALB) as a model enzyme to prepare lipase-GO aerogel (LGA). As shown in Fig. 1, we first mix the solutions of CALB and GO at pH 3.3 and shake it to form lipase-GO hydrogel. Then the lipase-GO aerogel is prepared by freeze-drying the hydrogel at −40 °C for 50 h. The hydrogel is

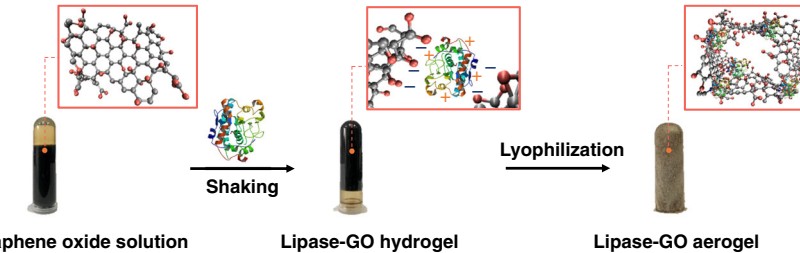

**Graphene oxide solution**        **Lipase-GO hydrogel**        **Lipase-GO aerogel**

**Fig. 1** Preparation procedures of LGA. LGA preparation by one-step co-gelation and lyophilization after mixing GO and CALB solutions

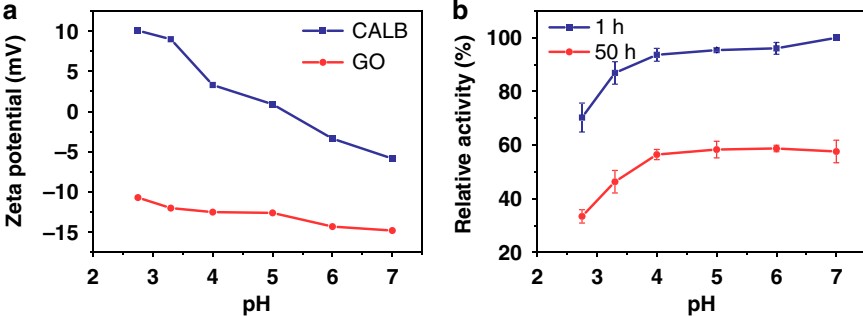

**Fig. 2** Properties of free CALB and GO for the optimization of pH in LGA preparation. **a** Zeta potentials of CALB and GO at various pH; **b** the relative activity of CALB after incubation at different pH for 1 h and 50 h. Error bars represent the s.d. of three replicates. Source data are provided as a Source Data file

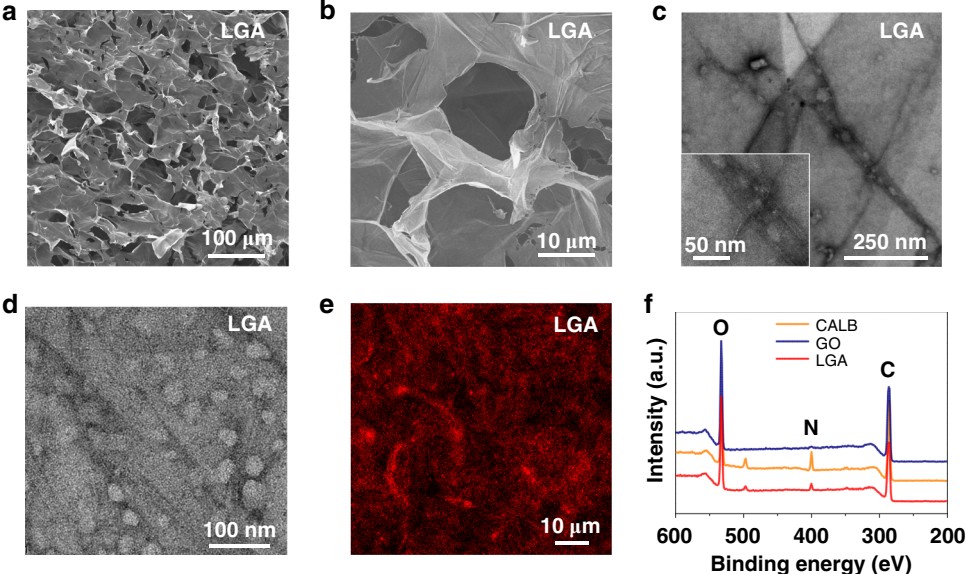

**Fig. 3** Structural characterization of LGA. **a**, **b** SEM images; **c** STEM image; **d** TEM images; **e** Laser confocal microscopy images with the CALB labeled with Rhb (red); **f** XPS data of CALB, GO, and LGA. Source data are provided as a Source Data file

formed through electrostatic interactions under pH 5 when the CALB molecules and GO sheets carry opposite charges (Fig. 2a). While a lower pH can facilitate hydrogel formation, it may also compromise the stability of CALB. To balance the opposite effects, we choose pH 3.3 for the preparation of LGA in the following experiments (Fig. 2b).

SEM images (Fig. 3a, b) reveal that LGA has a porous structure with a continuous distribution of the pore diameter between 10 to 100 μm. However, the naked GO sheets fail to form a stable porous structure after freeze-drying treatment (Supplementary Fig. 1a), probably due to the stacking of naked GO sheets after the removal of the entrapped water molecules. From the STEM and TEM images shown in Fig. 3c, d, we find that 80% of CALB ($3\,nm \times 4\,nm \times 5\,nm$) is deposited in the wrinkles of GO sheets, where the GO provides more available binding sites for CALB molecules. The confocal microscopy image (Fig. 3e) shows a molecular dispersion of CALB (labeled with Rhb) in the aerogel. According to the Brunauer–Emmett–Teller (BET) analysis (Supplementary Fig. 1b), the specific surface area of the LGA is $14\,m^2\,g^{-1}$ with meso- and macro- pores spreading all over the porous network. The carbon, nitrogen and oxygen contents are 58, 7, and 35% w.t., respectively, as interpreted from the X-ray photoelectron spectroscopy (XPS) results (Fig. 3f). All nitrogen

element in LGA should be from CALB, because GO is supposed to have no nitrogen atoms.

**Activity performance**. We examined the catalytic performance of LGA and lyophilized lipase powder for the esterification of geranyl acetate, a valuable natural perfume component[53], from geraniol and vinyl acetate at 20 °C under different water activities in a continuous reactor (Fig. 4a). As shown in Fig. 4b (the turnover number was the average reaction rate within the first two hours of reaction), LGA and lipase powder have the same optimal water activity of 0.34, while the apparent activity of LGA is 5 to 10-fold higher than that of lyophilized lipase powder at all water activities except $A_w = 0$ based on the same amount of lipase. It is noteworthy that LGA retains 67 % of its maximal activity under an absolutely anhydrous condition (i.e., $A_w = 0$). In contrast, the lyophilized lipase powder is completely inactive at the same condition. This may be attributed to the structural rigidity of enzymes in anhydrous conditions, which hindered the necessary conformational transition during catalysis, as detailed by Klibanov and his coworkers in their study of non-aqueous phase enzyme catalysis[1]. The drastic difference in enzyme activity indicates that lipase immobilized in aerogel retains its activity without relying on the $A_w$ in the gas phase.

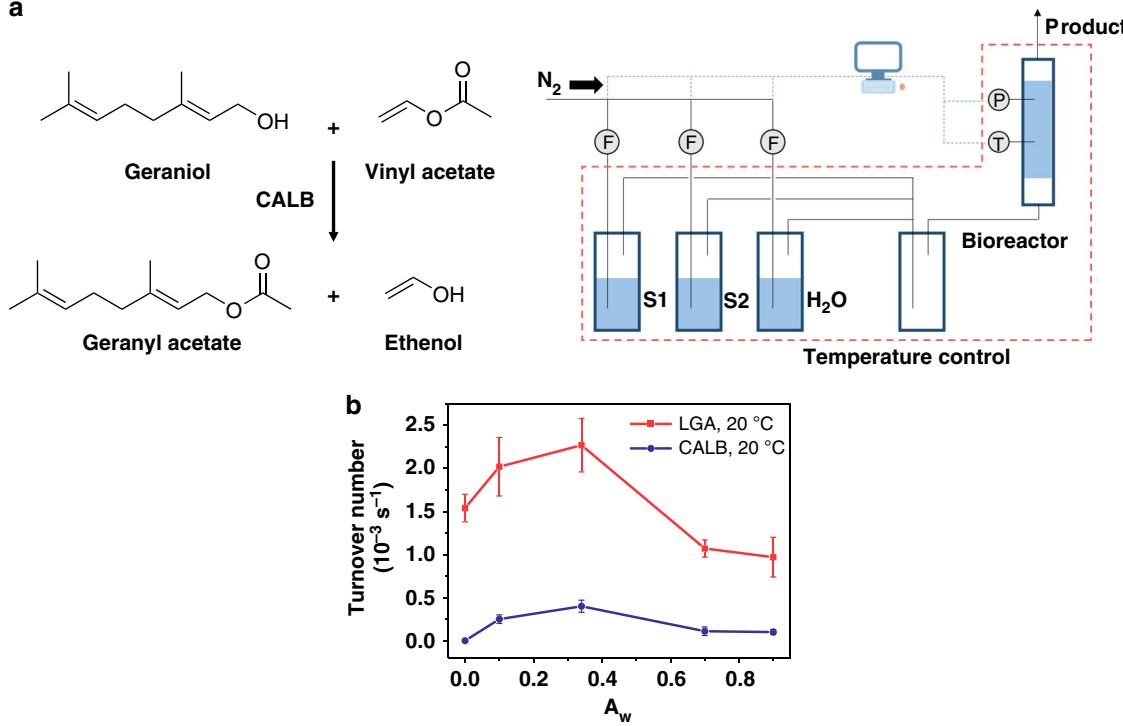

**Fig. 4** Experimental setup and activity test for transesterification. **a** Transesterification of geraniol and vinyl acetate catalyzed by CALB, and experimental setup for gaseous enzymatic catalysis. **b** the effects of water activity on the turnover numbers of LGA and CALB at 20 °C. Error bars represent the s.d. of three replicates. Source data are provided as a Source Data file

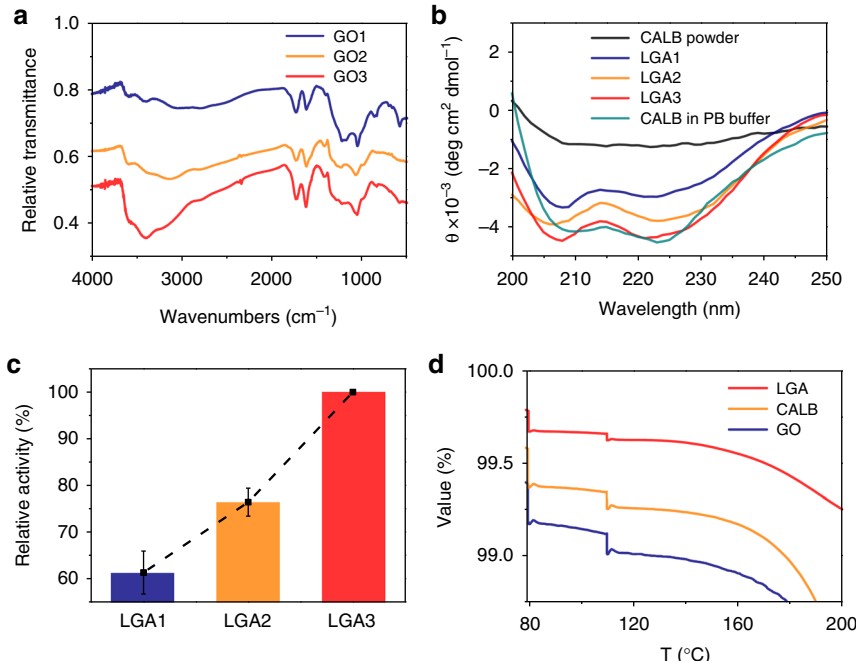

**Fig. 5** Characterization for mechanism demonstration. **a** FTIR spectra of GO1, GO2 and GO3; **b** solid-state circular dichroism spectra of dry CALB, LGA1, LGA2, LGA3, and CALB in a phosphate buffer; **c** relative activity of LGA1, LGA2, and LGA3; **d** TGA results for LGA3, CALB, and GO. Error bars represent the s.d. of three replicates. Source data are provided as a Source Data file

**Mechanism demonstration**. To find out the mechanism underpinning the high activity of LGA at $A_w = 0$, we employed solid-state circular dichroism (ssCD) spectroscopy to examine the secondary structures of CALB in the aerogel. We prepared three types of LGAs using GO sheets with the same oxygen content (35% w.t.) but different hydroxyl content (interpreted from the

infrared absorption of hydroxyl groups at wavenumbers from 3300 to 3600 cm$^{-1}$, Fig. 5a). The increase of hydroxyl groups in LGAs was confirmed by FTIR, which can be attributed to CALB (Supplementary Fig. 2). As shown in Fig. 5b, the dry lipase powder has almost lost its native secondary structure, while the lipase in different forms of LGAs (designated as LGA1, LGA2,

and LGA3) keeps a β-sheet content close to its native form in phosphate buffer and only undergoes some small changes in the content of α-helix. The increase in the hydroxyl content, i.e. from LGA1 to LGA3, helps maintain the secondary structures of the encapsulated lipase closer to its native form (Fig. 5b), as a result, the transesterification activity of LGA increases accordingly, as shown in Fig. 5c. These all suggest that the hydroxyl groups on the GO sheets play a vital role in the maintaining of the native conformations and activity of enzymes in an anhydrate gas phase.

To investigate whether the GO sheets stabilize the CALB structure by preserving water molecules or by directly interacting with the CALB, we applied the thermogravimetric analysis (TGA) to determine the water contents of the lyophilized LGA, CALB and GO, respectively. We determined the mass loss of the bounded water molecules by incubating these materials at 110 °C for 30 min[37]. As shown in Fig. 5d, the initial drops at 80 °C refer to the loss of free water, and the subsequent drops at 110 °C are attributed to the loss of bounded water. The amount of bounded water in LGA3 is determined to be 0.04%, which is much less than that in the lyophilized CALB (~0.08%) and GO (~0.1%). It indicates that the formation of LGA3 excludes significant amount of the bounded water molecules, and the water exclusion may be attributed to the formation of hydrogen bonds between CALB and GO. Therefore, we conclude that the hydroxyl-rich GO sheets direct interact with CALB by replacing a portion of bonded water molecules of lipase. In such a microenvironment, enzymes, like lipase, can maintain the native conformations and can undergo essential conformational changes for catalysis without relying on the presence of water molecules.

CALB has a short and flexible lid domain that favors the access to the active site when it exists in the open-lid state[54]. However, lipases from *Thermomyces lanuginosa*, *Bacillus thermocatenulatus*, and other bacteria, have a larger lid that can shield the active site when the lid is in the closed lid state[55,56]. Using molecular dynamics simulation, Yang and coworkers examined the effect of GO hydrophobicity on the lid state and enzyme activity for a lipase from *Alcaligenes sp.*[57]. They found that an increase in the hydrophobicity of GO helps open the lid and enhance the lipase activity. The hydrophobicity of GO can be conveniently tuned by its oxygen content, making GO-lipase aerogel available for various reactions. To further demonstrate the versatility of this method, *Candida rugosa* lipase (CRL) and *Porcine pancreatic* Lipase (PPL), two lipases with large lids, were tested for transesterification of geraniol and vinyl acetate. Supplementary Fig. 3a, b show that CRL-GO aerogel (CGA) and PPL-GO aerogel (PGA) are active at $A_w = 0$ while CRL and PPL powders have little activity. The secondary structures of CRL and PPL in powder are distinctively different from their counterparts in a soluble form, or in CGA and PGA complexes (Supplementary Fig. 3c, d). The changes in the secondary structures may explain the differences in their apparent activities.

**Stability measurements**. We examined the effects of water content and temperature on the stability of the LGA3 by incubation at temperatures ranging from 20 to 80 °C. Figure 6a shows that the LGA3 exhibits a maximum activity at $A_w = 0.34$ at all temperatures, and it can retain about 60% of the maximum activity at $A_w = 0$, while the lyophilized lipase powder is completely inactive. We carried out the reaction at $A_w = 0$ and 0.34 under different temperatures for 10 h (Fig. 6b–e). In all cases, the stability of the LGA3 is significantly improved compared with the lyophilized lipase powder, especially the LGA3 shows no apparent loss in activity over 10 h. The presence of $A_w$ in the gas flow ($A_w = 0.34$) can lead to the deactivation of lipase, and the deactivation will be accelerated if increasing the reaction temperature. The reduced

enzyme activity can be mainly attributed to the distortion of enzyme structures due to the unfavorable collision of free water molecules toward the enzymes and the thermal fluctuations of enzyme structures. We confirmed the denaturing of lipase in LGA3 from the structural changes after a 10-h operation at $A_w = 0.34$ and 80 °C by ssCD spectra (Supplementary Figure 4). In addition, the collapse and deformation of aerogel caused by the adsorption/desorption of moisture may bring additional mass transfer resistance (Supplementary Fig. 5). Finally, we examined the long-term performance of LGA3 at 20 °C and under anhydrous condition, i.e. $A_w = 0$, using an apparatus shown in Fig. 4a. The high activity of lipase-GO hydrogel in an anhydrous gas flux enables implementation of enzymatic reaction at room temperature, making the reaction environmentally more friendly, industrially more economical, and operationally more convenient. The products were analyzed by gas chromatography. As shown in Fig. 6f, the LGA3 retains 100% of activity over 500 h, which is, by far, the longest operating time ever reported. The extraordinary stability and activity of the enzyme confirm that the presence of GO resembles a natural microenvironment for enzymes.

In conclusion, we have established a facile method to engineer a water-like microenvironment for gas-phase enzymatic reactions by embedding enzymes in a hydrophilic, hydroxyl-rich GO aerogel matrix. The lipase-GO aerogel is capable to catalyze gaseous transesterification under anhydrous conditions. Characterizations including STEM, ssCD, and TGA and activity assays reveal that the GO stabilizes lipase by forming a water-like microenvironment which preserves the native structure of enzyme and thus the high activity in anhydrous catalysis. The excellent stability enables a long-duration continuous operation for 500 h. The ease of preparation and operation makes this robust enzyme-GO aerogel promising for industrial applications of gas-phase enzymatic catalysis.

## Methods

**Materials**. *Candida antarctica* Lipase B (product number L3170), *Candida rugosa* lipase (product number L1754), *Porcine pancreatic* Lipase (product number L3126), geraniol, vinyl acetate, geranyl acetate, 4-nitrophenyl butyrate, and Rhodamine B (Rhb) were purchased from Sigma-Aldrich. Graphite powder (325 mesh) was purchased from Qingdao Huatai Lubricant Sealing S&T Co., Ltd. (Qingdao, China). Other chemicals were all of analytical grade.

**Synthesis and purification of graphene oxide**. GO was prepared according to Hummers' protocol[58]. During a run, 0.5 g of NaNO₃ was added to a 250 mL flask containing 23 mL of H₂SO₄. The mixture was subjected to 5 min ultrasonication to completely solubilize the NaNO₃. The flask was then incubated in an ice bath with mechanical agitation at 150 rpm. Once the solution temperature was down to 0–5 °C, graphite powder of 1.0 g was added into the flask, followed by a slow addition of certain amount of KMnO₄ while the reaction temperature was maintained below 10 °C. Then the flask was incubated into to an oil bath at 35 °C and vigorously stirred (250 rpm) for 30 min. 50 mL of deionized water was slowly added to the flask while the temperature of oil bath was increased to 95 °C. After 15 min incubation, the mixture was poured into 500 mL of deionized water under agitation. H₂O₂ (5 mL 30%) was added dropwise until gas evolution was finished and the suspension color changed from dark brown to yellow.

The suspension was then filtered and washed with 300 mL HCl (3.7 wt.%) to remove metal ions. The obtained slurry was rinsed with deionized water to remove the residual acids. The slurry was redispersed in 500 mL of deionized water and further dialyzed against ultrapure water using a membrane tubing of MWCO of 8000–14,000 mol g⁻¹ for two weeks until the ionic conductivity of the dialysis water was below 5 μS cm⁻¹. The obtained GO aqueous dispersion was repeatedly centrifuged at 2000 r.p.m. for 20 min to remove non-exfoliated graphite particles, and then centrifuged at 10,000 rpm for 40 min to yield the GO stock suspension.

**Preparation of LGAs**. CALB purchased from sigma was in liquid form and was dialyzed against PB buffer (10 mM, pH 7) for 72 h at 4 °C using dialysis tubing with MW cutoff ranging from 8000–14000, followed by 48 h freeze-drying yielding CALB powder. During each run, 10 mg CALB powder was dissolved in 0.7 mL deionized water and then injected into 6.3 mL of 4.25 mg mL⁻¹ GO suspension at

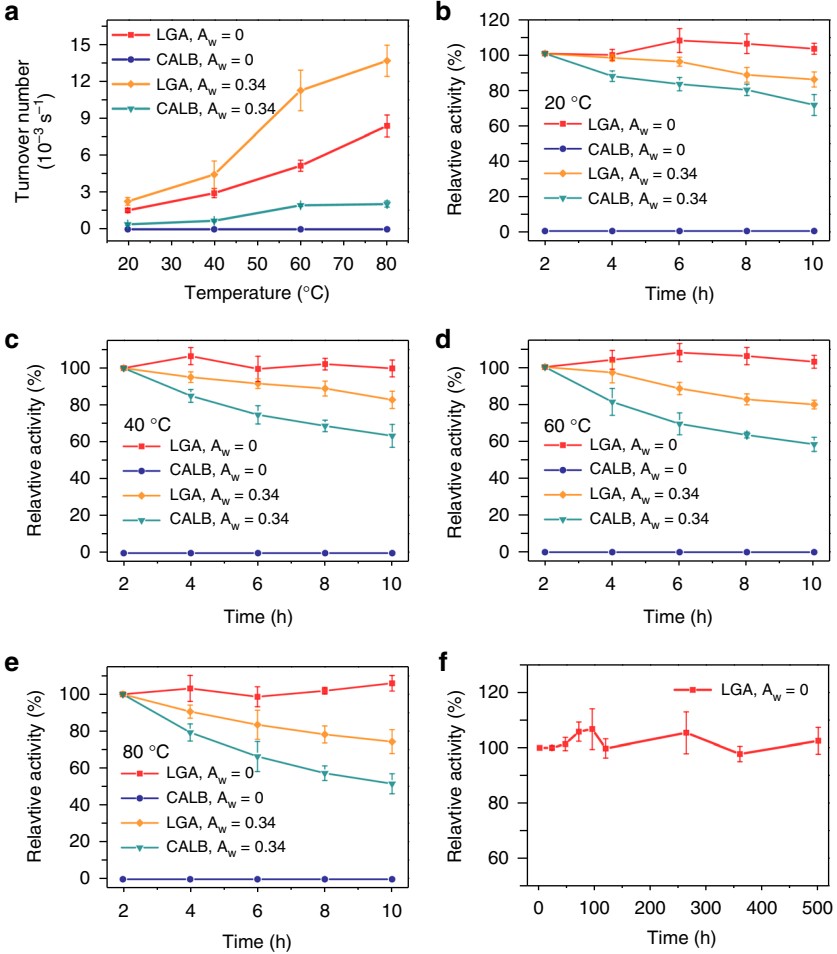

**Fig. 6** The long-term operation of LGA3 and CALB at different conditions. **a** The temperature dependence of turnover numbers of LGA3 and CALB at $A_w = 0$ and $A_w = 0.34$; **b–e** the thermostability of LGA3 and CALB at $A_w = 0$ and $A_w = 0.34$ at 20, 40, 60, and 80 °C; **f** the activity of LGA3 during a continuous operation for 500 h at 20 °C. Error bars represent the s.d. of three replicates. Source data are provided as a Source Data file

pH 3.3. After shaking fiercely for 20 s, the mixture was subjected to a 50 h lyophilization to form the LGAs.

**Enzymatic synthesis of geranyl acetate in the gas phase**. The apparatus is shown in Fig. 3a. Initially the storage tank 1 (S1) contains 20 mL of geraniol, and tank 2 (S2) contains 20 mL of vinyl acetate. In a typical run, the $N_2$ flows (10 mL min$^{-1}$) carrying the vapor of substrates from the tanks S1 and S2 were first mixed in a mixing tank and then were supplied into the reactor packed with LGAs or dry CALB powder. The product was collected in a 5 ml vial containing 2 mL of n-hexane placed in a 0 °C water bath. The product composition was analyzed by gas-phase chromatography (GC) as detailed in Supplementary Information. To study the catalytic performance in the presence of $H_2O$, we let the carrier gas purge through a tank filled with 20 mL liquid water. The water activity in the reactor was controlled by the $N_2$ flow rate.

**Labelling CALB with rhodamine B**. For the synthesis of rhodamine B-labelled CALB, 0.4 mL of 5 mg mL$^{-1}$ rhodamine B in DMSO solution was added into 4 mL of 5 mg mL$^{-1}$ CALB in a $Na_2CO_3$-$NaHCO_3$ buffer (50 mM, pH 9) and stirred for 12 h at 4 °C. After quenching the reaction with 50 mM $NH_4Cl$, the mixture was dialyzed against PBS (50 mM, pH 7) for 48 h at 4 °C to remove the unreacted dye.

**Circular dichroism spectroscopy**. Circular dichroism (CD) spectra of lipase in solution was determined on a Jasco-715 spectropolarimeter. For the enzyme in dry conditions, the circular dichroism spectra were examined by CD (Applied Photophysics Ltd Chirascan) with a continuous scanning mode at room temperature (23 ± 1 °C). A certain amount of lyophilized enzyme or immobilized enzyme was suspended in 5 mL pure water, and then the solution was flatted out on clean quartz slides by a dip-coating machine. The coated slides were dried completely before test. The spectra were recorded between 240 and 185 nm at a scan rate of 50 nm min$^{-1}$ with a resolution of 1 nm. And all spectra were corrected by subtracting the base line and averaged based on at least three scans for each sample.

**Gas-phase chromatography**. The concentration of substrates and products were analyzed by GC (Shimadzu GC-2010) equipped with a DB-5 column (30 m in length and 0.25 mm in diameter) and a FID detector. The helium flow rate was 3.5 mL min$^{-1}$ with a sample size of 0.3 μL and split ratio of 30:1. The temperature of injection port and detector were 270 °C and 260 °C respectively. The column was held at 150 °C for 2 min and then the temperature increased to 200 °C at 25 °C/min and then held for another 2 min. The result of every sample was an average of two tests within 5% error. The concentrations of chemicals were determined by external standard method, in which a series of different concentration of standard geraniol and vinyl acetate were tested before the analysis of substrates and products. The retention times for geraniol and vinyl acetate were 1.79 min and 2.45 min, respectively.

**Preparation of CGA and PPL**. During each run, 10 mg CRL/PPL was dissolved in 0.7 mL deionized water and then injected into 6.3 mL of 4.25 mg mL$^{-1}$ GO suspension at pH 3.3 (both enzymes are positively charged). After shaking fiercely for 20 s, the mixture was subjected to a 50 h lyophilization to form the CGA and PGA.

**Laser confocal microscopy**. Laser confocal microscopy was conducted on a Zeiss LSM-710 3-channel system with Plan-Apochromat 40×/0.95 Korr M27 objectives. The excitation wavelengths for CALB/RhB was 590 nm.

**Fourier transform infrared spectroscopy**. FTIR was performed on a Nicolet iS10 FT-IR Spectrometer.

**Thermogravimetric analysis**. TGA data was obtained on an Elementar vario EL III.

**Transmission electron microscopy**. TEM images were taken on a JEOL JEM-2010 high-resolution TEM with an accelerating voltage of 120 kV. The sample was stained with 1% sodium phosphotungstate solution at pH 7 before observation.

**Scanning electron microscopy**. SEM was conducted on a JEOL JSM 7401F field emission SEM with an accelerating voltage of 3.0 kV. The sample was sprayed with gold for 1 min.

## Data availability

All relevant data are available from the authors. The source data underlying Figs. 2, 3f, 4b, 5 and 6 and Supplementary Figs. 1b, 2, 3 and 4 are provided as a Source Data file.

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

## Acknowledgements

This work was supported by the National Natural Science Foundation of China under grant No. 21520102008. The authors are grateful to Mr. Mingmao Wu from the Department of Chemistry, Tsinghua University, for his generous help in the preparation of GO, Prof. Likun Yang from Xiamen University, for enzyme structure analysis, and Zhe Wang, for figure quality improvement.

## Author contributions

W.X. and Z.L. conceived the research. W.X., Y.Z., J.W., and Z.L. designed the experiments. W.X. performed all experiments. Z.F., Z.W., Y.J., G.C., G.J., and D.L. helped with aerogel preparation, enzyamtic catalysis, strctural and activy assays. W.X., Y.Z., J.W., and Z.L. wrote the manuscript. All the authors discussed the results and commented on the manuscript.

## Additional information

**Competing interests:** The authors declare no competing interests.

