## [Peer Review File · Nature Communications]

Reviewers' comments:

Reviewer #1 (Remarks to the Author):

In this manuscript, the authors fabricated lipase-GO aerogel and investigated the lipase activity and stability in a water-free gas flow. The idea is not novel enough, and there are several critical issues with the work. I do not recommend publication of the manuscript.

1. It seems that none of the experiment has been repeated. No error bars for all enzymatic activity analysis (Fig. 2b, 3c, and Fig. 4).
2. For lipase studies and applications, one of the key factors for catalysis is to open the lipase lid and maintain it in an open conformation to expose its active site, so called 'open lid'. This must be analyzed.
3. Lack adequate control. The lipase-GO aerogel was prepared at pH 3.3 and lyophilized right after. Other than GO, does pH plays a role in the activity and stability of lipase?
4. Literature not up to date. The idea is not novel enough, there have been previous reports regarding immobilization of lipase on GO to modulate its activity and stability.
5. Other than hydroxyl content, the oxygen content of GO is also an important effecting factor for interactions of GO with proteins, therefore should be analyzed as well.

Reviewer #2 (Remarks to the Author):

The present article describes a novel system for enzymatic reactions. The results are very interesting and should be published. My suggestions are for minor changes. I suggest the authors include the size of the CALB (either in kDa or nm) and also the pore size distributions (calculated by the adsorption equipment used by the authors) of the GO. This information will reinforce the statement that the CALB is probably not in the pores of the GO by rather in between crystals of the GO. It would be interesting to include also a N₂ adsorption isotherm and information of the GO prior to preparation of the LGA.

Line 108: I think it is Figure 1d and not 2d.

The authors write about GO with different OH contents but do not give any information on the contents of samples LGA1, LGA2 and LGA3. It would be interesting to include that information.

Line 213 to 214. ...storage tank 1 (S1) contains INITIALLY 20 mL of geraniol ...

Figure 4a shows TON for different samples. Missing information: TON calculated at what reaction time? As reaction rate decreases with time form all samples except LGA, $A_w=0$, it is important to mention the reaction time at which the TON was calculated.

Reviewer #3 (Remarks to the Author):

The submission by Liu et al reports a new method for immobilizing lipase (CALB) on graphene oxide (GO) with varied contents of hydroxyl group, resulting in a novel biocatalyst (LGA) which showed 5-10 times higher activity under a nearly anhydrous environment ($a_w \sim 0$) for a gas-phase transesterification of geraniol with vinyl acetate. The inspiring results were confirmed by comprehensively employing various techniques including ssCD and TGA, indicating that at least a part of the essential water molecules associated closely with the lipase protein could be replaced or

mimicked by the abundant hydroxyl groups on the GO sheet, to protect the native conformation of active lipase from being denatured under extremely anhydrous environment. More importantly, the GO-stabilized lipase preparation enables the long-term continuous biocatalytic synthesis of volatile chemicals in completely anhydrous gas flow.

I recommend to publish these interesting observations after proper modifications. If more enzymes especially lipases are tested, the results will be of greater importance. Actually, a simple activity assay is sufficient to compare the performance of different lipases immobilized onto the GO sheets.

Minor points:

1)Title: The phrase "an anhydrate gas flux" seems misleading since there is no hydrate or anhydrate compound in the gas flux except substrates/product. It may be changed to "an anhydrous gas flux". In addition, the "continuous" operation of enzymatic reaction is not necessarily enabled by GO or the GO-immobilized enzyme, since any forms of immobilized enzyme enable continuous reaction. Do the authors mean 'long-term' operation in this case?

2)Abstract: some phrases are incomplete or inexact, e.g., L22: 'solid circular dichroism' should read solid state circular dichroism? L24-25: 'water molecules...replaced by GO sheets' should better be changed to: 'water molecules...replaced by hydroxyl groups of GO sheets'.

3)The references need to be carefully formatted so as to adhere to the standard of the current journal.

4)Since the GO-lipase aerosol is extremely stable even operated at high temperatures up to 80 oC, why the long-term reaction was performed at so low a temperature as 20 oC? Please give a reasonable explanation.

Responses to Reviewer's Comments

Response to Reviewer #1

In this manuscript, the authors fabricated lipase-GO aerogel and investigated the lipase activity and stability in a water-free gas flow. The idea is not novel enough, and there are several critical issues with the work. I do not recommend publication of the manuscript.

Response:

We thank the reviewer for careful reading of this manuscript with constructive suggestions. One major challenge hindering enzymatic catalysis in gas phase is that enzymes exhibit ultra-low activity due to the increased rigidity in a dry state, while addition of moisture shortens the lifetime. This study presents, for the first time, an aerogel made from proteins and graphene oxide that can solve this paradox. By providing a hydroxyl-rich microenvironment, the aerogel can maintain the native conformations of enzymes meanwhile allow for essential conformational changes in water-free gas phase. The lipase-GO complex enables a 5 to 10-fold increase in the apparent activity in comparison to the lipase powder and maintains high activity for 500 hours in an anhydrous gas flux. As recognized by other reviewers, we believe that the novelty of this work, as well as its significance for enzymatic catalysis, merits publication of this manuscript.

1. It seems that none of the experiment has been repeated. No error bars for all enzymatic activity analysis (Fig. 2b, 3c, and Fig. 4).

Response:

Thanks for calling our attention about error bars. All results reported in this work were averaged over three parallel runs. In the revision, we presented the results with error bars.

2. For lipase studies and applications, one of the key factors for catalysis is to open the lipase lid and maintain it in an open conformation to expose its active site, so called 'open lid'. This must be analyzed.

Response:

Thanks for this insightful comment. The following discussion was added in the revised manuscript to analyze the lid effect on the apparent activity of lipase-GO:

“CALB has a short and flexible lid domain that favors the access to the active site when it exists in the “open-lid” state.⁵⁴ However, lipases from *Thermomyces lanuginosa*, *Bacillus thermocatenuatus* and other bacteria, have a large lid that can shield the active site when the lid is in the “closed-lid” state.^{55,56} Using molecular dynamics simulation, Yang and coworkers examined the effect of GO hydrophobicity on the lid state and enzyme activity for a lipase from *Alcaligenes sp.*⁵⁷ They found that increase in the hydrophobicity of GO helps open the lid and enhance the lipase activity. The hydrophobicity of GO can be conveniently tuned by its oxygen content, making GO-lipase aerogel available for various reactions.” (Page 8, line 16 from the bottom)

Moreover, we examined the effect of GO on the catalytic behavior of two additional lipases, *Candida rugosa* lipase (CRL) and *Porcine pancreatic* lipase (PPL). Both lipases have large lids and exhibit an enhanced catalytic performance in the lipase-GO aerogel. However, their native counterparts show little activity in the dry powder form. The new experiments further validate the versatility of the enzyme-GO complex for catalysis in an anhydrous gas flux.

“To further demonstrate the versatility of this method, *Candida rugosa* lipase (CRL) and *Porcine pancreatic* Lipase (PPL), two lipases with large lids, were tested for transesterification of geraniol and vinyl acetate. Figure S3a and S3b show that CRL-GO aerogel (CGA) and PPL-GO aerogel (PGA) are active at $A_w=0$ while CRL and PPL powders have little activity. The secondary structures of CRL and PPL in powder are distinctively different from their

counterparts in a soluble form, or in CGA and PGA complexes (Figure S3c and S3d). The changes in the secondary structures may explain the differences in their apparent activities.”
 (Page 8, line 9 from the bottom)

Figure S3. (a) Turnover numbers of CGA and CRL at 20 °C; (b) turnover numbers of PGA and PPL at 20 °C; (c) solid-state circular dichroism spectra of CGA, CRL powder and CRL in a phosphate buffer; (d) solid-state circular dichroism spectra of PGA, PPL powder and PPL in a phosphate buffer

3. Lack adequate control. The lipase-GO aerogel was prepared at pH 3.3 and lyophilized right after. Other than GO, does pH plays a role in the activity and stability of lipase?

Response:

We analyzed the pH effect on the activity and stability of CALB and presented the results as Figure S1 in the original submission. After 50h incubation, the relative activities are 47% and 55% at pH 3.3 and 4.0, respectively. To ensure stable gelation, we chose pH 3.3 for preparation of the lipase-GO aerogel. In the revision, we moved Figure S1 from Supporting Information to the main text (Figure 1) to address this important issue.

“The hydrogel is formed through electrostatic interactions under pH 5 when the CALB molecules and GO sheets carry opposite charges (Figure 1a). While a lower pH can facilitate hydrogel formation, it may also compromise the stability of CALB. To balance the opposite effects, we choose pH 3.3 for the preparation of LGA in the following experiments (Figure 1b).” (page 5, line 1)

4. Literature not up to date. The idea is not novel enough, there have been previous reports regarding immobilization of lipase on GO to modulate its activity and stability.

Response:

In the introduction portion of the revised manuscript, we provided a more comprehensive analysis of existing efforts to immobilize lipase on GO. To our best knowledge, all previous studies on GO-lipase complexes were concerned with enzyme activity and stability in liquid media. GO-lipase aerogel capable of long-term catalysis in anhydrous gas flux has not been reported before.

“Great efforts have been directed at immobilizing lipase on GO in liquid media. For example, Sofer and coworkers^{41, 42} studied lipases from *Rhizopus oryzae*, *Candida rugosa* and *Penicillium camemberti* and found that GO enhanced the thermal stability, solvent tolerance and activity of the enzyme in organic solvents. Husain and coworkers⁴⁷ attached lipase from *Aspergillus niger* onto polyaniline-coated silver-functionalized GO nanocomposites and achieved an improved thermal stability and activity in aqueous solutions. Improved stability in organic solvents was also reported for lipases from *Candida rugosa* and *Candida sp.* immobilization on modified GO sheets.^{43, 44, 45, 46} In addition, immobilization on GO-contained magnetic supports improves the stability of *Porcine pancreas* lipase,^{48, 49, 50} and *Candida rugosa* lipase⁵¹. ” (Page 4, line 15 from the bottom).

5. Other than hydroxyl content, the oxygen content of GO is also an important effecting factor for interactions of GO with proteins, therefore should be analyzed as well.

Response:

The oxygen content is 35% *w.t.* for all GO used in the present study. The value was determined by an elemental analyzer (Elementar Analysensysteme GmbH, vario EL III). This point has been clarified in the revised manuscript.

“We prepared three types of LGAs using GO sheets with the same oxygen content (35 % *w.t.*) but different hydroxyl contents (interpreted from the infrared absorption of hydroxyl groups at wavenumbers from 3300 to 3600 cm^{-1} , Figure 4a).” (page 7, line 10 from the bottom)

In the revised manuscript, we also measured the content of hydroxyl group of LGA 1-3 and the results are shown in Figure S2.

“The increase of hydroxyl groups in LGAs was confirmed by FTIR, which can be attributed to CALB (Figure S2).” (page 7, line 8 from the bottom)

Figure S2. FTIR spectra of LGA1, LGA2 and LGA3.

Response to Reviewer #2

The present article describes a novel system for enzymatic reactions. The results are very interesting and should be published. My suggestions are for minor changes. I suggest the authors include the size of the CALB (either in kDa or nm) and also the pore size distributions (calculated by the adsorption equipment used by the authors) of the GO. This information will reinforce the statement that the CALB is probably not in the pores of the GO by rather in between crystals of the GO. It would be interesting to include also a N₂ adsorption isotherm and information of the GO prior to preparation of the LGA.

Response:

First of all, the authors would like to thank the reviewer for pointing out the novelty and significance of this work for enzymatic catalysis. We also appreciate his/her constructive and detailed comments helpful for improving the quality of this manuscript.

We clarified that the size of CALB is about 3 nm × 4 nm × 5 nm. (page 6, line 5)

According to BET and SEM results, shown respectively in Figure S1b and Figure 2a, the lipase-contained aerogels have small pores, below 50 nm in diameter, and larger pores with the diameter ranging from 10 to 100 μm. The N₂ adsorption isotherm for freeze-dried GO fails to converge, suggesting that naked GO aerogel does not form a stable 3-D porous structure after the freeze-drying treatment. The lack of a stable porous structure may be attributed to the random stacking of naked GO sheets once the entrapped water molecules are removed. The same conclusion can also be drawn from the SEM results (Figure S1a). To a certain degree, CALB plays the role of molecular glue to connect GO sheets, resulting in a stable 3-D porous structure as shown in Figure 2. CALB molecules are intercalated between the GO sheets rather than inside the pores formed by GO sheets alone.

“SEM images (Figure 2a and 2b) reveal that LGA has a porous structure with a continuous distribution of the pore diameter between 10 to 100 μm. However, the naked GO sheets fail to

form a stable porous structure after freeze-drying treatment (Figure S1a), probably due to the stacking of naked GO sheets after the removal of the entrapped water molecules.” (page 6, line 1)

Figure S1. (a) The structure of freeze-dried GO; (b) the BET results of LGA.

1. Line 108: I think it is Figure 1d and not 2d.

Response: Thanks, this typo was corrected.

2. The authors write about GO with different OH contents but do not give any information on the contents of samples LGA1, LGA2 and LGA3. It would be interesting to include that information.

Response: Thanks for this insightful comment. In the revision, we measured the contents of hydroxyl groups in LGA 1-3 and the results are shown in Figure S2.

“The increase of the hydroxyl groups in LGAs was confirmed by FTIR, which can be attributed to CALB (Figure S2).” (page 7, line 8 from the bottom)

Figure S2. FTIR spectra of LGA1, LGA2 and LGA3.

3. Line 213 to 214. ...storage tank 1 (S1) contains *INITIALLY* 20 mL of geraniol ...

Response: Thanks. Corrections were made in the revised version.

“The apparatus is shown in Figure 3a. Initially the storage tank 1 (S1) contains 20 mL of geraniol, and tank 2 (S2) contains 20 mL of vinyl acetate.” (Page 11, line 14)

4. Figure 4a shows TON for different samples. *Missing information: TON calculated at what reaction time? As reaction rate decreases with time form all samples except LGA, $A_w=0$, it is important to mention the reaction time at which the TON was calculated.*

Response: Thanks. Throughout this paper, TON was calculated from the first two hours of reaction. The following note was added to clarify how TON was measured in this work:

“(the turnover number was the average reaction rate within the first two hours of reaction)”
(Page 6, line 3 from the bottom)

Response to Reviewer #3

The submission by Liu et al reports a new method for immobilizing lipase (CALB) on graphene oxide (GO) with varied contents of hydroxyl group, resulting in a novel biocatalyst (LGA) which showed 5-10 times higher activity under a nearly anhydrous environment ($a_w \sim 0$) for a gas-phase transesterification of geraniol with vinyl acetate. The inspiring results were confirmed by comprehensively employing various techniques including ssCD and TGA, indicating that at least a part of the essential water molecules associated closely with the lipase protein could be replaced or mimicked by the abundant hydroxyl groups on the GO sheet, to protect the native conformation of active lipase from being denatured under extremely anhydrous environment. More importantly, the GO-stabilized lipase preparation enables the long-term continuous biocatalytic synthesis of volatile chemicals in completely anhydrous gas flow.

I recommend to publish these interesting observations after proper modifications. If more enzymes especially lipases are tested, the results will be of greater importance. Actually, a simple activity assay is sufficient to compare the performance of different lipases immobilized onto the GO sheets.

Response:

The authors are grateful to the reviewer for pointing out of the novelty and importance of this work for enzymatic catalysis, particularly from the perspective of gas-phase synthesis. His/her constructive and detailed comments are also much appreciated.

Following the reviewer's suggestion, we tested the robustness of our method by applying it to two additional lipases:

“To further demonstrate the versatility of this method, *Candida rugosa* lipase (CRL) and *Porcine pancreatic* Lipase (PPL), two lipases with large lids, were tested for transesterification of geraniol and vinyl acetate. Figure S3a and S3b show that CRL-GO aerogel (CGA) and PPL-GO aerogel (PGA) are active at $A_w=0$ while CRL and PPL powders have little activity. The secondary structures of CRL and PPL in powder are distinctively different from their

counterparts in a soluble form, or in CGA and PGA complexes (Figure S3c and S3d). The changes in the secondary structures may explain the differences in their apparent activities.”
 (Page 8, line 9 from the bottom)

Figure S3. (a) Turnover numbers of CGA and CRL at 20 °C; (b) turnover numbers of PGA and PPL at 20 °C; (c) solid-state circular dichroism spectra of CGA, CRL powder and CRL in a phosphate buffer; (d) solid-state circular dichroism spectra of PGA, CRL powder and PPL in a phosphate buffer

1. Title: The phrase “an anhydrate gas flux” seems misleading since there is no hydrate or anhydrate compound in the gas flux except substrates/product. It may be changed to “an anhydrous gas flux”. In addition, the “continuous” operation of enzymatic reaction is not necessarily enabled by GO or the GO-immobilized enzyme, since any forms of immobilized enzyme enable continuous reaction. Do the authors mean ‘long-term’ operation in this case?

Response: Thanks. The title has been changed to

“Graphene Oxide Enabled Long-Term Enzymatic Transesterification in an Anhydrous Gas Flux”

2. Abstract: some phrases are incomplete or inexact, e.g., L22: ‘solid circular dichroism’

should read solid state circular dichroism? L24-25: 'water molecules...replaced by GO sheets' should better be changed to: 'water molecules...replaced by hydroxyl groups of GO sheets'.

Response: Thanks! We corrected those imprecise phrases.

“The solid-state circular dichroism measurement confirms that the lipase keeps its native conformation in the aerogel, and the thermogravimetric analysis shows a significant amount of water molecules essential for the lipase activity can be replaced by the hydroxyl groups of GO sheets.” (Page 2, line 7 from the bottom)

3. The references need to be carefully formatted so as to adhere to the standard of the current journal.

Response: Thanks, the references has been reformatted according to the standard of Nature Communication.

4. Since the GO-lipase aerosol is extremely stable even operated at high temperatures up to 80 °C, why the long-term reaction was performed at so low a temperature as 20 °C? Please give a reasonable explanation.

Response:

Conventional lipase catalysis in gas phase utilizes lipase powder and the reaction has to be conducted at high temperature in order to get high productivity. In other words, the apparent activity of lipase is extremely low in anhydrous gas phase. We demonstrated in the present study that lipase-GO aerogel has much higher activity, e.g., reaching 60% of the peak activity at the optimal water activity, i.e., $A_w = 0.34$, whereas the lipase power in anhydrous gas flux shows essentially no activity. The improved activity enables the enzymatic reaction at room temperature to achieve a high productivity, making the reaction more environmentally friendly, more economical, and more convenient to operate. We also tested the stability of lipase-GO hydrogel at 80 °C. In that case, the lipase-GO hydrogel shows no loss of activity for 10 hours

in an anhydrous gas flux. (Figure 5e).

The following sentences have been added to the revised manuscript:

“The high activity of lipase-GO hydrogel in an anhydrous gas flux enables implementation of enzymatic reaction at room temperature, making the reaction environmentally more friendly, industrially more economical, and operationally more convenient.” (Page 9, line 5 from the bottom)

REVIEWERS' COMMENTS:

Reviewer #1 (Remarks to the Author):

I am satisfied with the revisions and recommend to publish the manuscript.

Reviewer #2 (Remarks to the Author):

The authors submitted an original work about a novel enzymatic system. After reading the answers given to the referees I think the authors improved their manuscript and clarified any doubts about the work. I suggest the publication of the present work as is.

Reviewer #3 (Remarks to the Author):

The authors have addressed the questions that I was concerned with. Especially, I appreciate the the authors' efforts toward providing additional tests on the other two lipases (CRL & PPL), which indicates the generality and robustness of their methodology for activating lipases (not limited to CALB) under completely anhydrous environment by simply immobilizing them on GO-sheet. Now I think it is acceptable for publication.

Responses to Reviewer's Comments

Reviewer #1 (Remarks to the Author):

I am satisfied with the revisions and recommend to publish the manuscript.

Response:

Thanks.

Reviewer #2 (Remarks to the Author):

The authors submitted an original work about a novel enzymatic system. After reading the answers given to the referees I think the authors improved their manuscript and clarified any doubts about the work. I suggest the publication of the present work as is.

Response:

Thanks.

Reviewer #3 (Remarks to the Author):

The authors have addressed the questions that I was concerned with. Especially, I appreciate the the auhtors' efforts toward providing additional tests on the other two lipases (CRL & PPL), which indicates the generality and robustness of their methodology for activating lipases (not limited to CALB) under completely anhydrous environment by simply immobilizing them on GO-sheet.

Now I think it is acceptable for publication.

Response:

Thanks.